# All-optical temporal integration mediated by subwavelength heat antennas

Yi Zhang [1,4], Nikolaos Farmakidis [1,4], Ioannis Roumpos[2,4], Miltiadis Moralis-Pegios [3], Apostolos Tsakyridis [3], June Sang Lee[1], Bowei Dong [1], Yuhan He [1], Samarth Aggarwal [1], Nikos Pleros[3] ✉ & Harish Bhaskaran [1] ✉

Optical computing systems deliver unrivalled processing speeds for scalar operations. Yet, integrated implementations have been constrained to low-dimensional tensor operations that fall short of the vector dimensions required for modern artificial intelligence. We demonstrate an all-optical neuromorphic computing system based on time division multiplexing, capable of processing input vectors exceeding 250,000 elements within a unified framework. The platform harnesses optically driven thermo-optic modulation in standing wave optical fields, with titanium nano-antennas functioning as wavelength-selective absorbers. Counterintuitively, the thermal time dynamics of the system enable simultaneous time integration of ultra-fast (50 GHz) signals and the application of programmable, non-linear activation functions, entirely within the optical domain. This unified framework constitutes a leap towards large-scale photonic computing that satisfies the dimensional requirements of AI workloads.

Matrix–vector operations underpin most artificial intelligence (AI) algorithms, yet remain the Achilles' heel of computing hardware—consuming disproportionally large energy and taking a long time to run[1]. Linear optical circuits have gained attention for their potential to mitigate these challenges. Implemented in artificial neural networks (ANNs) they can reduce the latency of multiply-accumulate (MAC) operations and improve the overall energy efficiency through low-loss optical interconnects. Optical accelerators map machine learning operations directly in hardware by spatially multiplexing guided optical modes and weighting them via phase[2–6] or amplitude modulation[7–11]. A particular strength of these systems is their ability to perform massively parallel operations by multiplexing in space[3–5], wavelength[7–10,12–18] and mode[19,20]. This inherent advantage of photonics has been exploited to achieve PetaOPS-scale computing[21], demonstrating an impressive 2 order of magnitude improvement compared to state-of-the-art electronics.

However, the large physical footprints occupied by optical unitary cells and the insertion losses inherent in both active and passive components have made scaling beyond 64×64 weight matrices challenging[22]. Efforts to overcome these limitations by tiling photonic subassemblies[23], matrix factorization[24], and weight pruning[25] offer partial solutions but remain far from achieving the scale required for the billions of parameters implemented in complex AI tasks. Temporal multiplexing can be used to increase the parameter space and maintain a small system footprint[26–30], while interleaving both wavelength and time can be used for tasks including real-time video recognition[31,32]. Temporal multiplexing for advanced AI computing have also been demonstrated in recent photonic AI accelerator architectures, which integrate time, wavelength, and space multiplexing to achieve increased computational power[33,34]. Yet, a fully optical solution for large-vector processing—capable of in situ accumulation and nonlinear activation—remains

[1]Department of Materials, University of Oxford, Parks Road, Oxford, UK. [2]Department of Physics, Aristotle University of Thessaloniki, Thessaloniki, Greece. [3]Department of Informatics, Aristotle University of Thessaloniki, Thessaloniki, Greece. [4]These authors contributed equally: Yi Zhang, Nikolaos Farmakidis, Ioannis Roumpos. ✉e-mail: npleros@csd.auth.gr; harish.bhaskaran@materials.ox.ac.uk

unrealized. Temporal accumulation of weighted optical signals has only been realised in the electronic domain with photoreceiver charge accumulation[35] while inter-layer analogue non-linearities are performed either offline with additional electronic circuitry[4,35] or through opto-electronic and subsequent electro-optic conversions[12,13,36,37].

Here we propose an optically end-to-end framework for large vector processing based on the concept of a photonic-heater-in-lightpath (PHIL) unit, as illustrated in Fig. 1a. Multiplexed optical signals $X_i(t)$ with discrete wavelengths ($\lambda = 1, 2, 3 \ldots i$) are weighted using cascaded modulators which perform the element-wise multiplication $A_i(t) = X_i(t) \cdot w_i(t)$ at a time $t$ directly in the optical domain, and this

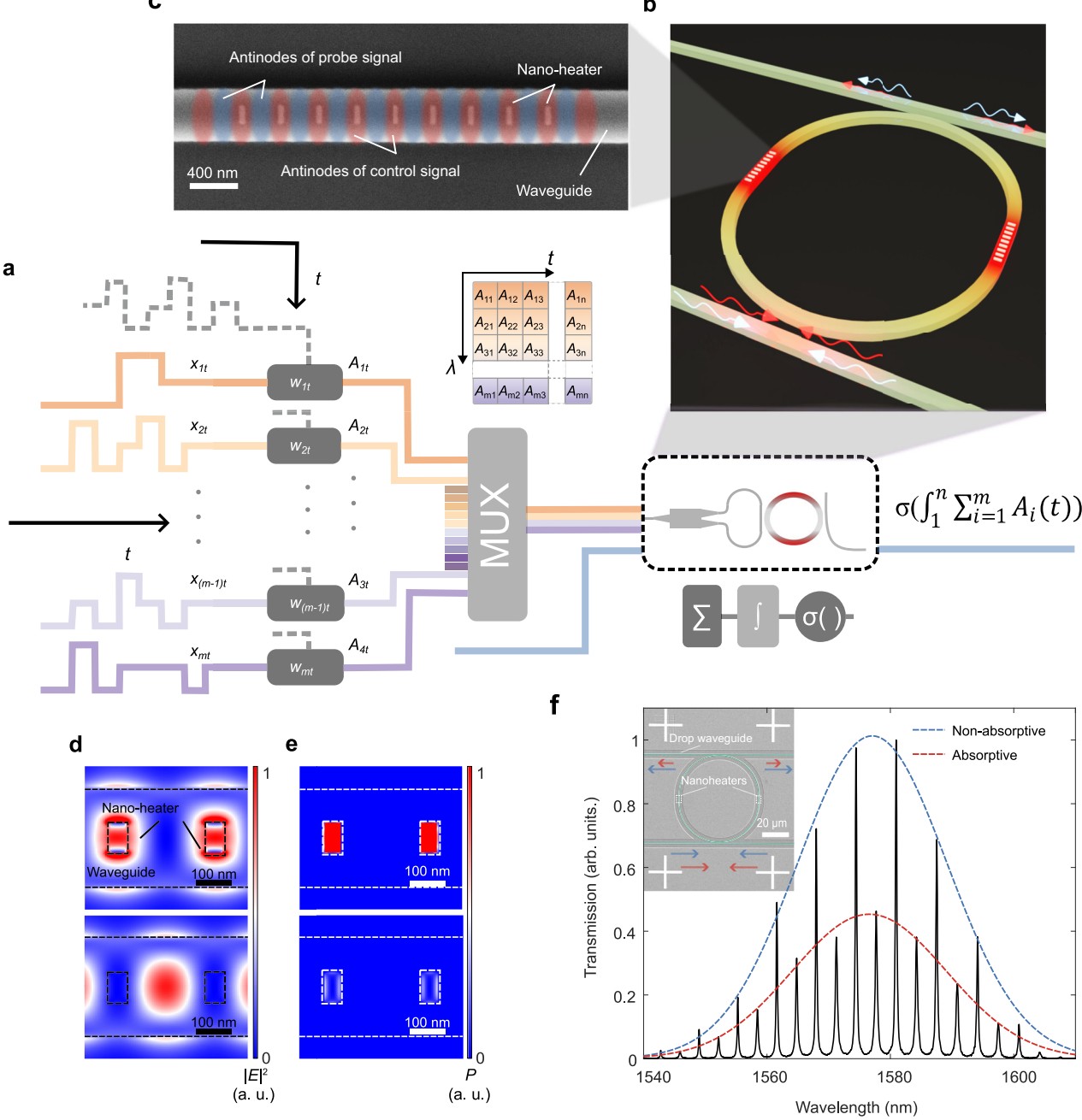

**Fig. 1 | Photonic-heater-in-lightpath: Photon manipulation via the Opto-Thermal-Optic (O-T-O) effect. a** A platform for large-vector processing. Input vectors are wavelength-time (λ-t) encoded and accumulated simultaneously in PHIL. Leaky time integration of multiple control signals (CS) is performed across wavelengths and encoded onto a new optical carrier (probe signal, PS) at a new wavelength. **b** Control signals (CS, red arrow) which carry the weighted input signals, and probe signals (PS, blue arrow) are coupled to the micro ring resonator (MRR) from both directions. The clockwise and counterclockwise travelling waves produce a standing wave pattern within the ring resonator. **c** Fabricated photonic integrator: Scanning electron micrograph of titanium nano-heaters on a ring integrator. Absorptive (non-electrical, fully optical) photonic-heater-in-light-path (PHIL) are designed to spatially match the anti-nodes of CS while overlapping with the nodes of PS. 9 nano-heaters are deposited on both left and right side of the ring at a pitch of 308 nm. **d** Normalized E-field of two antiphase wavelengths at the locations of the PHIL antennas. **e** Normalised power absorbed per unit area by nano-heaters indicating the presence of lossy and near-lossless wavelengths. **f** Experimentally measured transmission spectrum of the device (inset) showing periodic lossy and low-loss resonances. The blue line is a Gaussian fit to the grating coupler spectrum, while the red line includes the loss due to the PHIL antennas on ring.

large optical vector $A_i(t)$ is mapped in both wavelength and time scale to be processed simultaneously in a single PHIL unit. The transient optical signals are subsequently coupled into the PHIL (Fig. 1b), forming standing wave patterns with spatially periodic intensity variations. The intensity maxima are engineered to overlap with nano-heaters (Fig. 1c) positioned atop the waveguide, enabling efficient absorption and conversion of the incident optical energy into heat. Simultaneously, a low-power continuous-wave optical probe is injected into the same waveguide to read out the result of the MAC operation by quantifying a phase-shift imparted by the thermo-optic effect of the carrier waveguide (Full device microscope image shown in Supplementary Fig. 1). As the intensity minima of the probe align with the heater regions, it traverses the structure with minimal optical loss. 50-GHz time-multiplexed signals are optically accumulated by exploiting the MHz time dynamics of the heat dissipation, performing a leaky time integration of the signal, with the ability to integrate up to 6500 weighted inputs at each wavelength. Meanwhile, reconfigurable nonlinearities are directly applied to the accumulated signal by leveraging the spectral shape of high-Q resonances. Importantly in the proposed scheme, nonlinearly activated data is encoded onto a newly generated optical carrier facilitating optical cascadability in photonic neural networks[38].

## Results

### PHIL: Realisation of lossy and lossless spectral bands

To enable an all-optical reconfigurable unit with minimal insertion loss, we engineer photonic integrated circuits that exhibit distinct lossy and near-lossless spectral bands. The former is employed for modulation (control signals, CS) and the latter supports low-loss readout and facilitates cascadability (probe signals, PS). The underlying mechanism hinges on the creation of a photonic standing wave through the interference of two counter-propagating coherent optical waves. The field distribution of such a system is described by a periodically varying optical field in space[39–41] with spatially-fixed nodes (locations of zero electric field) and antinodes (locations of maximum electric field) at any arbitrary point in time ($t$). The spatial periodicity of these is derived by the superposition of the forward and counter-propagating waves in space ($x$) and time ($t$) as:

$$
\begin{aligned}
E(x,t) &= E_0 \sin\left(\frac{2\pi x}{\lambda} + \omega t + \Delta\varphi\right) + E_0 \sin\left(\frac{2\pi x}{\lambda} - \omega t\right) \\
&= 2E_0 \sin\left(\frac{2\pi x}{\lambda} + \frac{\Delta\varphi}{2}\right) \cos\left(\omega t + \frac{\Delta\varphi}{2}\right)
\end{aligned}
\tag{1}
$$

where $E_0$ is the amplitude of the input waves, $\Delta\varphi$ is the phase difference between two coherent inputs.

Assuming $\Delta\varphi = 0$, positions along the propagating direction that satisfy even multiples of a quarter wavelength

$$
x = 2n \cdot \frac{\lambda}{4}
\tag{2}
$$

where $n = (..., -3, -2, -1, 0, 1, 2, 3, ...)$ form nodes (where the amplitude is zero) while odd multiples of a quarter wavelength

$$
x = (2n+1) \cdot \frac{\lambda}{4} \quad n = (\cdots, -3, -2, -1, 0, 1, 2, 3, \cdots)
\tag{3}
$$

form anti-nodes (where the amplitude is maximal).

Crucially for the work here, two wavelengths $\lambda_1$ and $\lambda_2$ can be chosen such that they are nearly perfectly out of phase with the anti-nodes of one wavelength coinciding with the nodes of the other and vice versa. This effect is illustrated using finite difference time domain simulations (FDTD, *Lumerical solutions*) in Fig. 1d. For two antiphase

wavelengths $\lambda_1$ and $\lambda_2$ the field at the absorber shows a maximum for $\lambda_1$ and a minimum for $\lambda_2$.

In this case, placing a nanoscale absorber at the location of an antinode for $\lambda_1$ induces strong attenuation at $\lambda_1$ whilst $\lambda_2$ is nearly unperturbed, limited only by the finite size of the absorber and the positioning of the absorber. This effect is illustrated using finite difference time domain simulations (Fig. 1e) (FDTD, *Lumerical solutions*) where with the total power absorbed by $\lambda_1$ is 17 times the amount absorbed at $\lambda_2$ (Supplementary Fig. 2). The combined effects of the standing wave formation and the absorptive nanoantenna thus generate periodic absorptive and transparent spectral bands.

We implement this concept on ring-resonators (Fig. 1b) in order to i) discretize the coupled wavelengths and ii) to amplify this effect by coupling to high quality resonances and exploit the spectral non-linearity in the modulation result. To create interference between incoming signals, a multimode interferometer (MMI) is employed to split the optical signal into two paths which is then coupled to the micro ring resonator (MRR) from opposite directions. The clockwise and counter-clockwise travelling waves produce a standing wave pattern within the ring resonator with varied spatial periodicity of $\frac{\lambda}{2n_{eff}}$, and outcoupled to the drop port where the transmission change is measured.

To couple light into the ring, the perimeter ($L$) of the ring must be an integer multiple of the input light wavelength:

$$
L = \frac{m\lambda}{n_{eff}}
\tag{4}
$$

where m denotes the mode number, corresponding to the number of resonance wavelengths supported within the MRR cavity. These modes primarily fall into two categories of whispering-gallery-mode (WGM) resonances, odd and even modes, which appear alternately at successive resonant wavelengths. From Eq. (2) and (3) it stands that, these two groups of resonance modes will form out-of-phase standing waves at midpoint of the MRR $\left(x = \pm \frac{L}{4} = \pm m \cdot \frac{\lambda}{4n_{eff}}\right)$ [40].

An array of optical nano-heaters, made of titanium which is both absorptive and has a high melting point, are strategically placed at locations $x = \pm \frac{L}{4}$ (middle left and right part of the ring) to absorb at the even modes. The centroidal separations of the nano-heaters is 308 nm corresponding to $\frac{\lambda}{2n_{eff}}$. Figure 1c shows a zoomed scanning electron micrograph (SEM) of these absorbers. Light coupled to the device is split into two counter-propagating waves. The clockwise and counterclockwise waves interfere to form a standing-wave field, with nanoheaters positioned at the antinodes for localized optical absorption. The out-coupled signals are symmetrically distributed between the two output ports (Supplementary Fig. 1). The spectral response of the device (Fig. 1f) shows typical periodic transmission peaks of an add-drop ring resonator yet with the creation of alternating low- and high-quality factor resonances corresponding to the engineered lossy and lossless wavelengths.

### Optical end-to-end encoding across wavelengths

We next evaluate the ability of the absorptive modes to induce resonance shifts within the micro-ring resonator and thereby modulate other co-propagating optical signals. Optical power is coupled in the absorptive bands (i.e., wavelengths that correspond to antinode spacing of a chosen set of wavelengths) and is converted into heat. The heat generated leaks to the guiding structure inducing a broadband thermo-optic shift. This allows the amplitude of a control signal (CS) to be imprinted onto a spectrally distant probe signal (PS) via the thermo-optic effect.

Figure 2a illustrates the concept of optical end-to-end encoding, where amplitude-modulated control signals ($\lambda_{CS}$) induce a spectral shift in the resonator that modifies the transmission of a weak probe signal ($\lambda_{PS}$) at a near-lossless wavelength. Figure 2b shows the

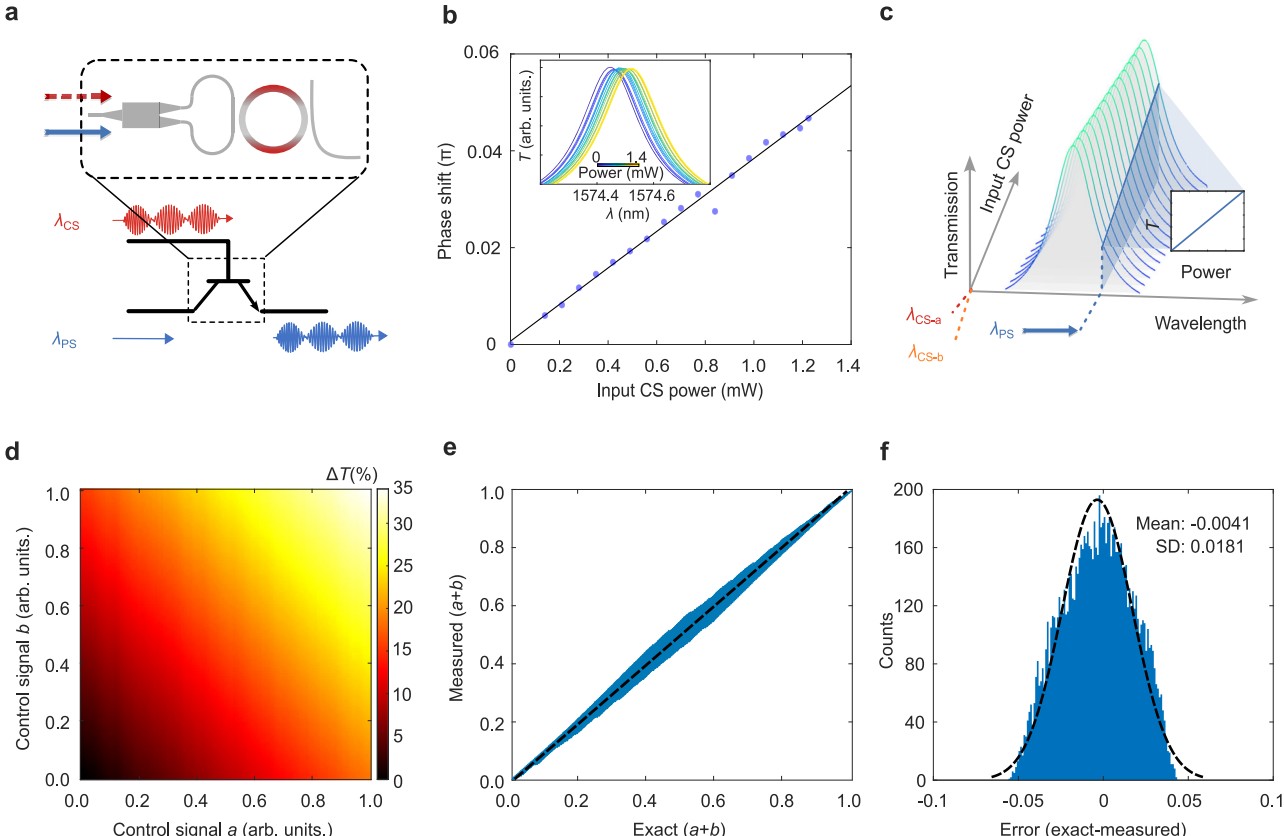

**Fig. 2 | Incoherent optical end-to-end encoding. a** Concept of optical end-to-end encoding. The control optical power ($\lambda_{CS}$, red) is accumulated in heat and shifts the ring spectrum via the thermo-optic effect. The total power is then encoded in the amplitude of a probe wavelength ($\lambda_{PS}$, blue). **b** The spectrum shifts linearly with increasing power of the control signal. (Inset: Spectral response collected at non-absorptive mode while increasing input power at 1583.84 nm.) **c** Concept of incoherent optical end-to-end encoding, where incoherent summation of different control signals ($\lambda_{CS-a}$ and $\lambda_{CS-b}$) is achieved in the form of heat. **d** Linear addition of signals carried by the intensity of two incoherent CS to the PS. **e** Measured addition results compared with the exact value and corresponding linear fit. **f** Error of the addition operation and corresponding normal fit.

experimental validation of the scheme in Fig. 2a where we quantify the shift of the spectrum of the ring resonator as a function of the input power at the control signal $\lambda_{CS}$. A linear relationship is observed, with a sensitivity of 0.04 π/mW when the CS is applied at an absorptive resonance. This responsivity remains highly uniform (5.3% variation) across devices with identical design parameters (Supplementary Fig. 3). When the probe and control wavelengths are swapped—placing the CS at a lossless wavelength—the shift drops significantly to 0.0087 π/mW (Supplementary Fig. 4). This further confirms the wavelength selectivity of the mechanism that the spectral response is only modulated by the absorptive modes and the low-optical loss at non-absorptive modes of the device proposed above.

Beyond single-wavelength modulation, this concept extends naturally to incoherent optical summation across multiple absorptive control signals. Figure 2c presents the principle, in which two distinct control signals $\lambda_{CS-a}$ and $\lambda_{CS-b}$ at different wavelengths induce cumulative heating within the same resonator. These signals are added incoherently through thermal accumulation and transduced by a shared probe signal located at a low-loss wavelength.

Experimental results in Fig. 2d demonstrate this behaviour: the optical transmission of the probe signal reflects the total power carried by both $\lambda_{CS-a}$ and $\lambda_{CS-b}$. Figure 2e shows excellent agreement between measured outputs and theoretical predictions of the summation. The error distribution in Fig. 2f is narrow and well-fit by a Gaussian distribution, indicating high linearity and low noise in the optical addition process.

This thermal accumulation scheme is scalable: additional absorptive control signals can be added across the spectral range of the resonator, each contributing to the overall phase shift experienced by the probe. The experimental setup and signal preparation for these tests are detailed in Supplementary Fig. 5 and Supplementary Fig. 6. The experimental evaluation of the system scalability and quantification of the analogue precision under multi-wavelength channels is further shown in Supplementary Fig. 7.

## Optical time integration of 50-GHz signals

We now demonstrate time-domain integration of high-speed optical signals by exploiting the inherent thermal response time of PHIL. Supplementary Fig. 8 and Supplementary Fig. 9 characterize the temporal dynamics during heating and cooling cycles, revealing a leaky integration time constant of ~130 ns. Within this time window, optical signals arriving at absorptive wavelengths are thermally accumulated, enabling analog integration in the time domain.

Figure 3a illustrates the concept: control signals (CS) modulated at 50 GHz (i.e., 20 ps pulses) are sent in bursts lasting 20 ns, separated by 1 μs intervals to allow full thermal recovery between integration events. These signals are injected into the device at an absorptive wavelength (1556.4 nm), with the experimental setup detailed in the Methods and Supplementary Fig. 10.

Figure 3b, d show the applied pulse patterns: a baseline sequence of alternating ones and zeros, and three randomized sequences (Random 1–3) with the same total number of high and low bits (500

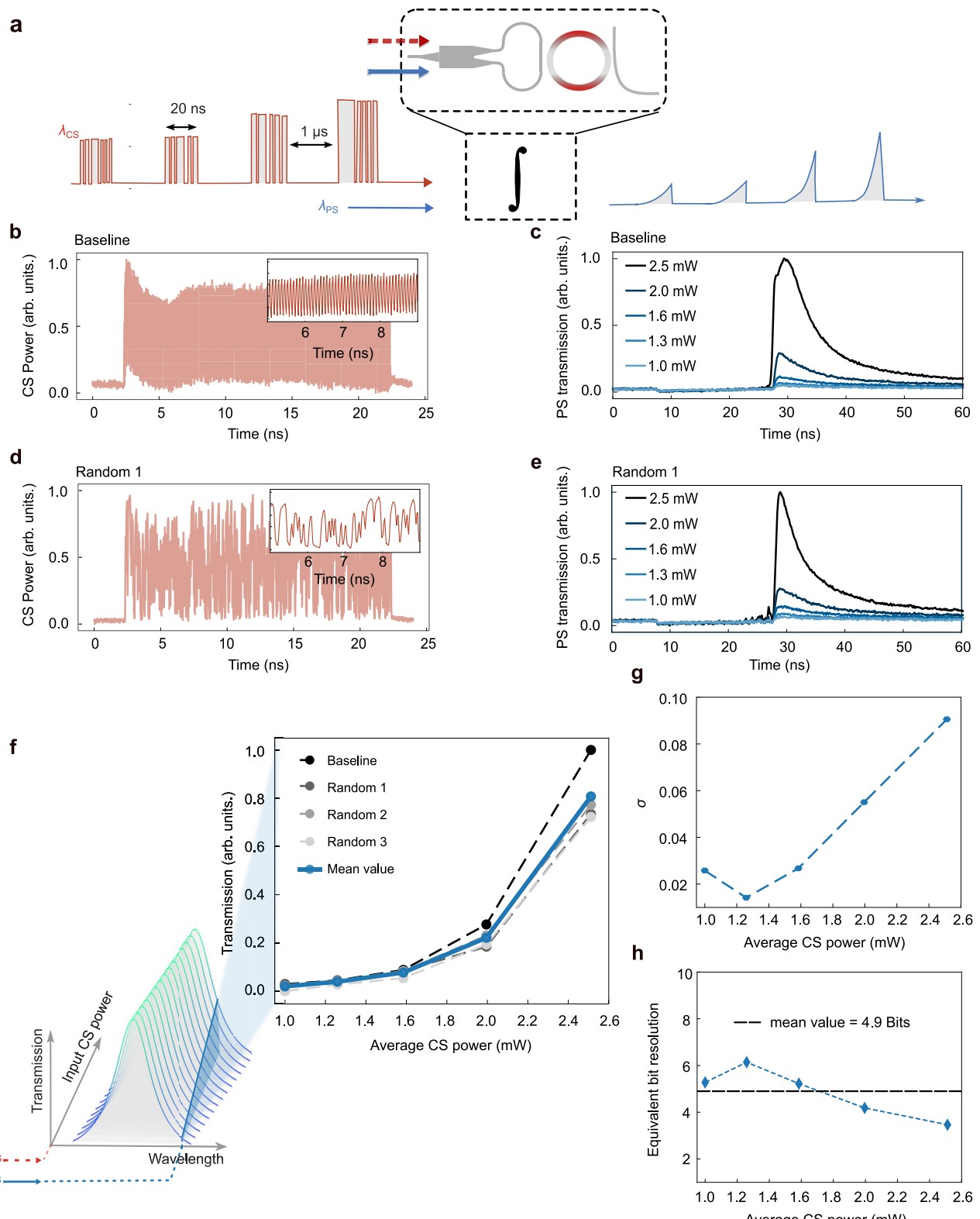

**Fig. 3 | Accumulation of 50 GHz signals over time. a** Concept of time integration on optical signals **b, d** Different CS pulse trains at 50 GHz, maintaining the same total number of zeros and ones (500 ones and 500 zeros). **c, e** Optical response of PS at relative CS pulse trains with average controlling power. **f** Optical power of control signals is accumulated over time by heat, with the spectrum shifted by the overall thermal-optic effect. Integration peak values at certain PS in response to varied input CS power, which shows a ReLU-like nonlinear transfer function due to the intrinsic Lorentzian spectrum of MRR. **g** Standard deviation of errors of integration peak values from the mean value at different input power. **h** Dynamic bit resolution for different CS input powers as BR = $\log_2(1/\sigma)$. Near 5-bit resolution is achieved, which is sufficient for AI applications.

ones and 500 zeros), ensuring identical average energy. The corresponding optical responses of the probe signal (PS), measured at a fixed spectral position beyond the resonance, are shown in the Fig. 3c, e. As the CS bursts are absorbed and converted into heat, the induced thermo-optic shift accumulates and modulates the PS. The peak of PS response denotes the resultant value. After the burst, the integrated value starts to decay over time due to heat dissipation.

Figure 3f quantifies the integration peak for various CS average powers, indicating a nonlinear mapping due to the Lorentzian lineshape of the micro-ring resonator. The response exhibits a ReLU-like transfer function, which is inherently useful for neuromorphic processing. To assess its computational relevance, the experimentally measured activation function was implemented in a simulated MNIST digit-recognition task (Supplementary Fig. 11). The PHIL-based activation achieved classification accuracy comparable to that obtained with the conventional ReLU function, confirming its suitability for analog optical neural computation.

To evaluate the precision of this integration, we compute the standard deviation of the integration peaks across multiple trials. As shown in Fig. 3g, the variability is minimal. The dynamic bit resolution for different CS input powers is defined as:

$$\text{Bit Resolution(BR)} = \log_2\left(\frac{1}{\sigma}\right)$$

where $\sigma$ is the standard deviation of the errors from the mean value. Figure 3h shows that the PHIL achieves ~5-bit dynamic resolution across CS power levels—sufficient for AI applications such as classification and inference[42].

This approach enables all-optical time integration of vectors up to size 1000 within a 20 ns window, with nonlinear transformation applied during accumulation. Counterintuitively, given the 130 ns integration window, up to 6500 high-speed (50 GHz) signals can be integrated per wavelength with this MHz-scale time dynamics. When scaled to 40 WDM channels with a comb source, this architecture supports fully optical accumulation of over 250,000 weighted inputs—a performance milestone for scalable neuromorphic photonics.

### All-optically reconfigurable activation functions

While previous sections demonstrated that the resonance shift of the MRR scales linearly with CS power, we now show that the shape of the resonance itself can be exploited to implement programmable nonlinear activation functions entirely in the optical domain—eliminating the need for external circuitry, a known bottleneck in photonic accelerators.

As illustrated in Fig. 4a, by tuning the probe wavelength along the Lorentzian profile of an MRR resonance, different nonlinear transfer functions can be applied to the PS in response to accumulated CS-induced heating, depending on the relative position of the probe wavelength to the resonance peak.

To experimentally characterize these effects, we inject a continuous-wave CS at the centre of an absorptive resonance and gradually ( ~ Hz) modulate its power over multiple cycles (Supplementary Fig. 12) using a variable optical attenuator (VOA). The PS is probed at three wavelengths: (i) 1587.26 nm (resonance centre), (ii) 1587.28 nm (slightly right of centre), and (iii) 1587.30 nm (further right of centre), as shown in Fig. 4b–d.

(i) At 1587.26 nm (Fig. 4e), the PS transmission exhibits a nonlinear decrease as CS power increases—characteristic of a Lorentzian activation function. Similar transfer functions have been employed to compensate for optical fibre nonlinearity in long-haul communications[43].

(ii) At 1587.28 nm (Fig. 4f), the PS response initially decreases, then increases—indicating the spectral shift has crossed the resonance centre.

(iii) At 1587.30 nm (Fig. 4g), the PS response shows a saturating nonlinear increase with CS power—resembling sigmoid or bounded activation functions used in photonic neural networks[11].

The modulation trace remained highly consistent across probe wavelengths, with a standard deviation below ±1.7 % during one-hour continuous operation (Supplementary Fig. 13), confirming the excellent thermal and spectral stability of the PHIL device in open-loop conditions.

These non-linear activation functions are then successfully applied to the across-wavelength summation results in the all-optical domain (Supplementary Fig. 14). These results confirm that a single photonic unit can support multiple nonlinearities by simply tuning the probe wavelength—without altering the device structure or extra electrical circuits. We further demonstrate that these programmable transfer functions can be applied to time-integrated signals, as shown in Fig. 5a.

In this experiment, 20 GHz CS pulse trains (200 ones and 200 zeros, 20 ns total duration) are sent to the PHIL at absorptive wavelengths, with PS signals probed at three distinct wavelengths corresponding to different spectral slopes. Figure 5b-d show the normalized PS responses to different CS power. As before, the accumulated thermal shift modulates the PS transmission, and the spectral position dictates the activation shape.

Figure 5e-g show the integration peaks for different input sequences (same average energy), captured at the three PS wavelengths, highlighting the extracted nonlinearity which aligns with previous analyses. The spectral position further to the right of the resonance exhibits a ReLU-like nonlinear response, which could be beneficial for neural network applications.

These results confirm the device's ability to serve as a nonlinear optical integrator, applying reconfigurable activation functions during accumulation. This capability is especially beneficial for neuromorphic processing, where nonlinearities must be tunable and scalable across layers

## Discussion

We demonstrate a photonic-heater-in-lightpath architecture that enables fully optical temporal integration and reconfigurable nonlinear processing by leveraging spatially engineered standing wave patterns for selective thermal modulation. This design exploits standing-wave-induced field localization to achieve wavelength-dependent optical absorption at nanoscale absorbers precisely aligned with field antinodes. Control signals at distinct wavelengths are integrated directly in the optical domain, with the accumulated result encoded onto a carrier probe with reconfigurable nonlinearities—eliminating the need for intermediate electro-optic conversions, a key limitation in scalable photonic computing.

Counterintuitively, this architecture harnesses the inherently slow dynamics of the thermo-optic effect—typically considered a bottleneck—to integrate optical signals modulated at 50 GHz. The resulting leaky integrator has the potential to accumulate temporal multiplexed inputs over a 130 ns window, enabling the processing of up to 6500 input signals per wavelength. When combined with 40-channel WDM, this corresponds to over 250,000 accumulated values per inference cycle. This approach reframes slow photothermal effects as powerful analog computing mechanisms and provides a compact, scalable, and reconfigurable platform for high-speed signal processing and neuromorphic computing—paving the way for photonic systems capable of matching the scale and complexity of modern AI workloads.

## Methods

### Device Fabrication

Silicon on insulator wafers (220 nm Si on 3 μm oxide) were used as substrates for integrated photonic circuit. Substrates were cleaned in

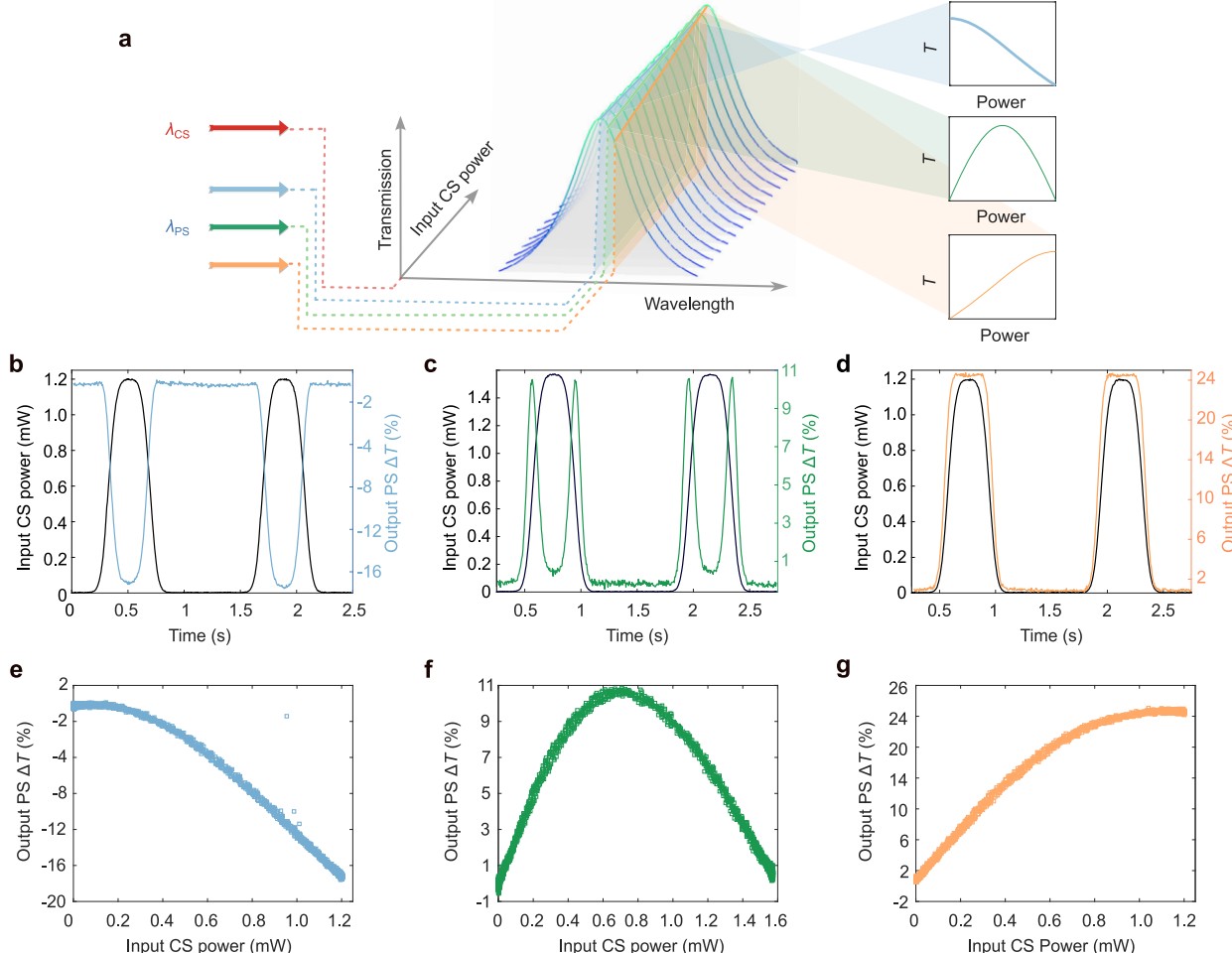

**Fig. 4 | Activation function engineering across wavelengths. a** Schematic of non-linear modulation with Lorentz shape of MRR spectrum. At different probe wavelengths (PS), different non-linear operations can be applied to the PS in response to input control signal (CS) power, while input PS power is kept constant. **b**–**d** Measured optical response at different probe wavelength at certain CS. Probe wavelength shifts from left to the right at the same absorptive wavelength. The different colours in the PS traces denote responses measured at distinct probe wavelengths. Here a variable optical attenuator (VOA) was employed to control the CS power. $\Delta T = (T - T_0)/T_0$, is the change in transmission of the level $T$ with respect to the baseline $T_0$, and $T_0$ varies with the choosing PS. **e**–**g** Non-linear functions obtained from different probe wavelength in a single all-optical unit.

acetone and IPA, dried using nitrogen, and subsequently baked at 150 °C to remove water condensation. The samples were subsequently spin-coated with CSAR65 positive (electron-beam lithography) EBL resist and baked at 150 °C for 3 minutes to remove all solvents from the thin film. Samples were patterned in a JEOL JBX5500 system. The samples were finally etched at a depth of 110 nm by reactive-ion etching (RIE, Oxford Instruments) with a gas mixture of CHF3/Ar/O2 to create the waveguides, grating couplers and ring resonators. A second EBL was carried to define the pattern of antennas, and the process was the same as the last run. After the development, the patterned chip was deposited with 20-nm titanium by electron-beam evaporation to form nano-heaters.

**Measurement setup**

The incoherent addition measurement was carried out using the setup described in Supplementary Fig. 5. A tuneable laser (Santec, TSL-550) was used to optically probe the device at non-absorptive (even-mode-resonant) wavelengths and the probe signal was sent as continuous wave into the PIC through the 10% port of a 90/10 coupler, with the polarization of the incident light adjusted by a polarization controller, and other two laser sources (N7711A, Keysight) were used to send addition information. Variable optical attenuators (Thorlabs, V1550)

were used to encode data in the optical power of control lines. Output from the device was filtered by (OTF-320, from Santec), and only the transmission change in probe line is measured after device. A circulator was used to protect instruments by blocking reflection from chip. Throughout the experiment, the power of the control signal was monitored through the 1% port of the 99/1 coupler.

Control signals a and b were carried by distinct absorptive (odd-mode-resonant) wavelengths and combined using a 50/50 coupler. The combined signals were routed through the 90% port of a 90/10 coupler before being coupled into the PIC. Two variable optical attenuators (Thorlabs, V1550) encoded data into the optical power of the control lines by varying transmission, which was regulated by the voltage from the DAQ via computer control. After characterizing the attenuators, we assigned 101 evenly spaced power steps to each control signal by adjusting the voltage from 0-5 V. As illustrated in Supplementary Fig. 6, during the measurement, Control signal a increased by one power step every 101 time steps, while Control signal b increased by one power step per time step. This setup allowed us to measure and analyze 10,201 addition events.

The time-scale response characterization was carried out using the setup described in Supplementary Fig. 8. A tuneable laser (Santec, TSL-550) was used as probe line and the probe signal was sent into the

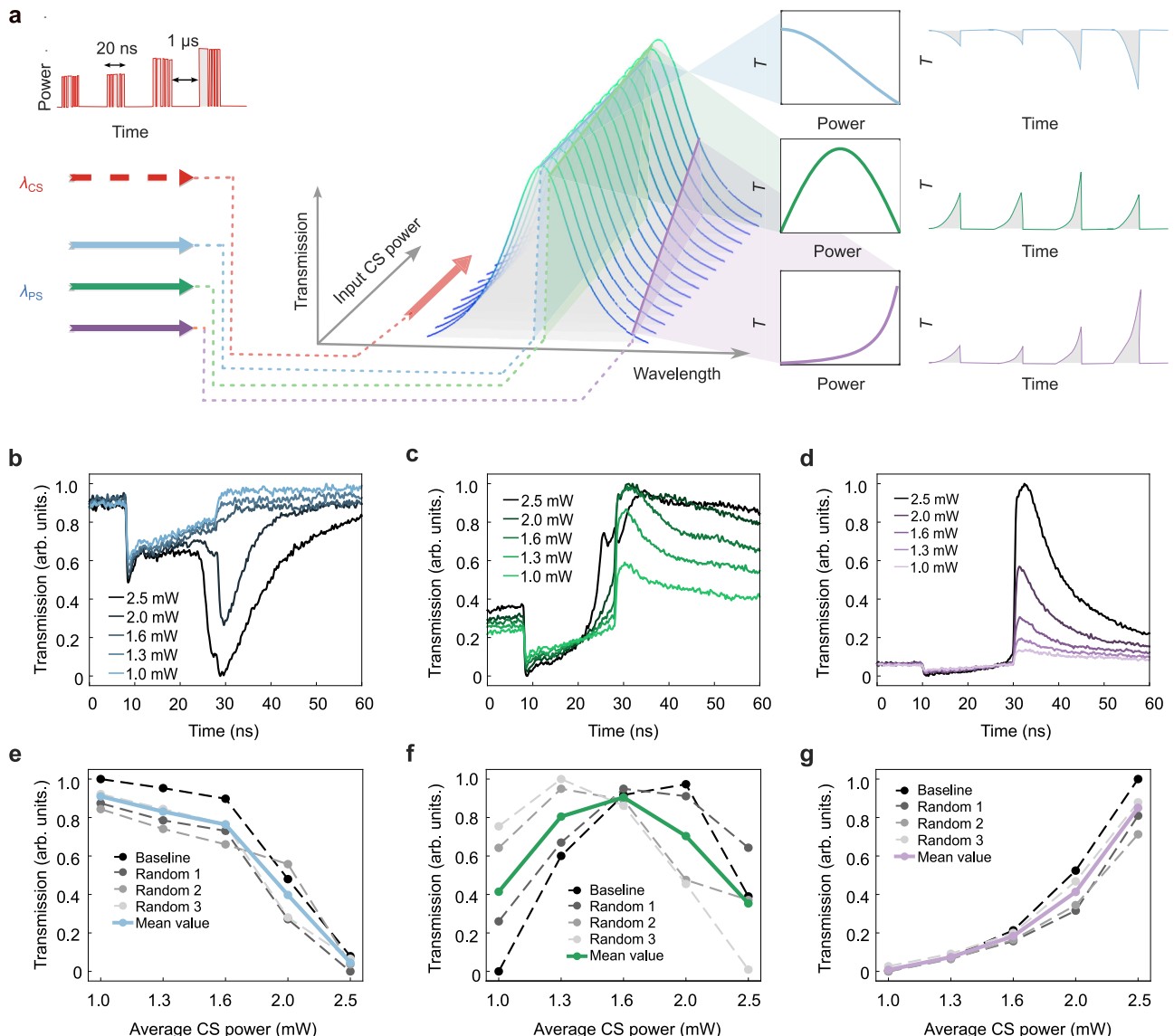

**Fig. 5 | Time integration with optically programmable activation functions.**
**a** Schematic of all-optically applying programmable non-linear activation functions to the integration results within the same optical unit. At different probe wavelengths (PS), different non-linear operations can be performed on the integration results and imprinted to the transmission of PS, while input PS power is kept constant. **b**–**d** Normalized optical response at different probe wavelength after 20-ns pulse trains at certain control signals (CS). The different colours in the PS traces denote responses measured at distinct probe wavelengths. **e**–**g** Integration peak values at certain PS in response to different CS series. Non-linear functions are obtained from different probe wavelength in a single all-optical unit due to the intrinsic Lorentzian spectrum of MRR.

PIC as a continuous wave through the 10% port of a 90/10 coupler, with the polarization of the incident light adjusted by a polarization controller. A circulator was used to protect instruments by blocking reflection from chip. The wavelength of the probe signal was chosen at the half maximum of an odd-mode peak to obtain linear response to overall input power during phase shifting. Another laser source (Santec, TSL-570) was used as control line to shift the spectrum all-optically. The incident power of the control signal was adjusted by an electro-optical modulator (EOM). (Lucent, 2623NA), which controlled by an electrical pulse generator (AFG3151C, Tektronix). The control signal was sent into the PIC as a continuous wave via the 99% port of a 99/1 coupler, followed by the 90% port of a 90/10 coupler. The polarization of the incident light was adjusted using a polarization controller. Throughout the experiment, the power of the control signal was monitored through the 1% port of the 99/1 coupler. Rectangular control pulses are sent into the device with different pulse widths

(from 10 ns to 500 ns). Output from the device was filtered by (OTF-320, from Santec) and amplified by an erbium-doped fibre amplifier (EDFA) to compensate for chip losses. Before collected by the photo-diode, the second filter is applied to remove the EDFA amplified spontaneous emission (ASE) noises.

The high-speed measurements were conducted using the experimental setup shown in Supplementary Fig. 10. A tuneable laser source (Santec TSL-550) was used for the probe signal, while another laser source (ANDO-TSL AQ4321A) was used for the control signal. The control signal was launched into a Mach-Zehnder modulator (MZM) with a 40 GHz bandwidth, after its polarization was adjusted. The MZM was driven by an AWG (Keysight 8194 A) to generate the ps pulses. The electrical output signal from the AWG was amplified by an RF amplifier (SHF L806A). The optical output was then amplified by an EDFA, and an optical bandpass filter (OBPF) with a 1 nm bandwidth was used to reject ASE) noise. The amplified signal was launched into a second

MZM to generate the 20 ns envelopes. This MZM was driven by an AWG (ARB RIDER AWG-4081). The output optical signal was amplified again with an EDFA to achieve the required input power in the PIC, and the incident power of the control signal was adjusted by a variable optical attenuator. The trace and optical power of the input signal were monitored by an optical power meter (Santec MPM-210) and a digital oscilloscope (Keysight N1000A). At the output, an OBPF with a 1 nm bandwidth blocked the control signal, allowing only the probe signal to be monitored. Finally, an EDFA amplified the output to compensate for chip losses, and the probe signal was captured by a 70 GHz bandwidth photodiode and recorded on another channel of the digital oscilloscope.

## Data availability

All data supporting the findings of this study are provided within the paper and its Supplementary Information. The source data are available on Figshare under the identifier (https://doi.org/10.6084/m9.figshare.30651389).

## Code availability

The code used in the present work is available from the authors upon request.

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

## Acknowledgements

The authors acknowledge support from the European Union's Horizon Europe research and innovation programme under grant agreements No. 101098717, (HYBRAIN), No. 101017237 (PHOENICS) and No 101098717 (RESPITE). Y.Z. and H.B. acknowledge financial support from EPSRC.

## Author contributions

Y.Z., N.F. and I.R. contributed equally to this project. Y.Z. carried out fabrication and simulations, and together with N.F. and I.R. set-up experiments. H.B., N.P., and N.F. conceived the concept and design, contributed to the experiments and analysis, and provided the structure for the project. M.M.P., A.T., J.S.L., Y.H., B.D., and S.A. contributed to the experiments and data collection. Y.Z., N.F. and H.B wrote the manuscript with substantial contributions from all authors. All authors provided in-depth discussions and suggestions at all stages of the work and discussed the results.

## Competing interests

H.B is currently employed at a semiconductor company although this work was done during his time at the University of Oxford. He also owns shares in other photonic device firms and has either been granted or has applied for several related patents in the field. The remaining authors declare no competing interests.
