## [Transparent Peer Review file · Nature Communications]

All-Optical Temporal Integration Mediated by Subwavelength Heat Antennas

Corresponding Author: Professor Harish Bhaskaran

Version 0:

Reviewer comments:

Reviewer #1

(Remarks to the Author)

Authors have demonstrated an all-optical neuromorphic computing system based on time division multiplexing, capable of processing input vectors exceeding 250,000 elements within a unified framework. The overall results are interesting to the entire community. The manuscript can be considered for publication in Nature Communications after the following remarks are fully addressed.

1. Authors need to provide SEM images and dimensions of the entire matrix in Fig. 1, rather than just schematics and subsections of nanometers.
2. Authors are required to provide uniformity information regarding OTO effect across different matrix units, such as wavelength tuning linearity for different devices, or power consumption for different devices.
3. What is the overall insertion loss for such a design, which can be a crucial parameter for any potential all-optical applications?
4. Have authors evaluated the thermal cross-talk issues of nano heaters, especially for large scale integration?
5. Is there feedback control for MRRs? Do authors observe any wavelength drifting over time during CS power sweeping in Fig. 4?
6. Would it be possible to implement CS pulse trains at higher speeds above 100GHz?

Reviewer #2

(Remarks to the Author)

This paper reports an all-optical computing platform that performs time integration and nonlinear processing using subwavelength thermal nano-heaters embedded in a photonic circuit. The system's ability to apply reconfigurable activation functions is experimentally validated by leveraging thermo-optic effects and standing-wave field patterns. The manuscript can be accepted for publication provided the following questions are adequately addressed:

- 1) Performance of the system should be benchmarked against other all-optical technologies in terms of energy efficiency, speed, and precision.
- 2) Thermal crosstalk between neighboring components is not quantified, which could negatively impact performance. The authors should discuss potential mitigation strategies and scalability constraints.
- 3) The nonlinear activation functions depend on probe wavelength alignment with Lorentzian resonance line shapes. How sensitive is the activation function to fabrication variability or wavelength drift, e.g., due to temperature fluctuations?

Reviewer #3

(Remarks to the Author)

The manuscript by Zhang et al. presents a novel and compelling approach to all-optical temporal integration and nonlinear activation using a photonic-heater-in-lightpath (PHIL) architecture based on a silicon microring resonator with subwavelength titanium antennas. The work leverages wavelength-selective absorption and thermo-optic effects to achieve simultaneous accumulation and programmable activation, demonstrating potential for scalable neuromorphic photonic computing. The concept is innovative, the device fabrication and experimental validation are thorough, and the results are promising. However, to firmly establish the practical viability and scalability of this approach for real-world AI workloads, the following points require further clarification and quantitative support.

1. The energy consumption per operation is a critical metric for neuromorphic hardware. Please provide the normalized on-chip energy per accumulated MAC operation at a representative vector length. This should include the energy cost of both the control and probe signals, as well as any amplification or conditioning required.
2. How does the effective number of bits scale with an increasing number of simultaneous wavelength channels, and what is the primary source of noise or crosstalk that limits the precision in a WDM scenario?
3. Can a multi-stage cascade be demonstrated using only optical probe signals for interconnects, and what is the required per-stage link budget and optical signal-to-noise ratio to ensure the nonlinear activation function remains stable under the cumulative effect of thermal loading and resonance shifts?
4. How do the fundamental constraints imposed by the Kramers-Kronig relations between absorption and phase shift limit the maximum achievable differential responsivity without introducing unacceptable levels of optical loss or distortion for the probe wavelength?
5. Can the authors demonstrate the device performing an end-to-end neuromorphic inference task such as MNIST handwritten digit recognition, and report how the achieved accuracy compares against a state-of-the-art electronic baseline?

Version 1:

Reviewer comments:

Reviewer #1

(Remarks to the Author)

Authors have addressed all my concerns in the revised manuscript. Current version meets the scope of NC.

Reviewer #2

(Remarks to the Author)

The authors have addressed all my comments adequately. The manuscript is now ready for publication.

Reviewer #3

(Remarks to the Author)

The authors have made substantial improvements in response to the previous comments. The revised manuscript addresses all concerns and is suitable for publication in its current form.

We thank the reviewers for their excellent feedback to our work. We have addressed the points raised by the reviewers which have considerably strengthened the manuscript. Please find our point-by-point response in blue and corresponding changes made in the manuscript here in red and highlighted in the manuscript.

Reviewer #1:

Authors have demonstrated an all-optical neuromorphic computing system based on time division multiplexing, capable of processing input vectors exceeding 250,000 elements within a unified framework. The overall results are interesting to the entire community. The manuscript can be considered to be published in Nature Communications after following remarks are fully addressed.

We thank the reviewer for their positive and constructive feedback and their recommendation to publish. Below we provide point-by-point responses and outline the corresponding revisions to the manuscript.

1. Authors need to provide SEM images and dimensions of entire matrix in Fig. 1, rather than just schematics and subsection of nanometres.

We thank the reviewer for the feedback and for pointing out the omission of important geometrical parameters. We have included an SEM image of the fabricated device in Figure 1. To further clarify the structure, we have included an annotated optical micrograph and zoomed SEM of the device with all relevant dimensions in supplementary figure. S1. As the reviewer has pointed out, the system is evaluated with off-chip multiplexing and weighting while the on-chip PHIL unit (ring +

Fig. S1: Optical microscope image and SEM image of PHIL-loaded ring resonator

subwavelength heaters) are operated on-chip which is shown in the updated electron micrographs.

As per the reviewer's suggestion, we have included the projected architecture of the components integrated on-chip in supplementary section S10:

Supplementary Section S10:

As shown in Fig. S1, the PHIL architecture consists of a racetrack microring resonator (radius: 30 μm) connected via a compact 1:2 MMI splitter and waveguide S-bends. The thermo-optic phase-shifter

arms used were intentionally elongated for tuning experiments and are not essential to the core PHIL function, and can be replaced with compact directional bends in the final system.

We base our area estimation on the actual fabricated layouts from our experimental devices, supported by optical microscope and SEM images.

PHIL unit:

The layout footprint includes:

1. A 30 μm -radius microring: $A_{\text{ring}} = \sim 0.003 \text{ mm}^2$.
2. A 1:2 MMI splitter (75–90 μm in length, 5.5–6 μm in width): $A_{\text{MMI}} = \sim 0.0005 \text{ mm}^2$.
3. Routing and coupling bends occupying the remainder of the layout footprint.

Accounting for fabrication spacing rules and routing clearance, the overall footprint of one PHIL unit is estimated at:

$A_{\text{PHIL}} = 0.009 \text{ mm}^2$ (conservative bounds: 0.008–0.010 mm^2).

This estimate is consistent with standard IMEC PDK and similar silicon photonics foundries. All area and throughput calculations in the main text are based on this conservative figure.

Modulator array estimation:

To ground footprint projections in realistic foundry components, we assume applying Si-Ge electro-absorption modulators (EAMs) operating at $\approx 50 \text{ Gbaud}^1$, each with device length 170 μm and effective width 5 μm . For both input encoding and weighting, we assume using two cascaded EAMs per wavelength channel. The per-EAM area is $170 \mu\text{m} \times 5 \mu\text{m} = 850 \mu\text{m}^2 = 0.00085 \text{ mm}^2$. Thus, with an estimated gap at 5 μm between two cascaded EAMs, the two-EAM cascade occupies around 0.001725 mm^2 before routing/contacts. Including conservative overhead for electrodes, vias, and bends, we budget 0.0022–0.0026 mm^2 per wavelength channel.

1. Per-EAM area: $A_{\text{EAM}} = 170 \mu\text{m} \times 5 \mu\text{m} = 850 \mu\text{m}^2 = 0.00085 \text{ mm}^2$
2. Two-EAM cascade (encode + weight): $A_{2\text{EAM}} = 2 \times 0.00085 = 0.0017 \text{ mm}^2$
3. Layout overhead (electrodes/contacts/bends): 0.0022–0.0026 mm^2 per channel.

With one PHIL shared across N_λ wavelengths, the tile area is $A_{\text{PHIL}} + N_\lambda A_{\text{ch}}$ with $A_{\text{PHIL}} = 0.009 \text{ mm}^2$ and $A_{\text{ch}} = 0.0024 \text{ mm}^2$ (ranges 0.008–0.0100 and 0.0022–0.0026 mm^2).

Die-area examples:

$$40\text{-WDM channels: } A \approx 40 \times 0.0024 + 0.009 \approx 0.105 \text{ mm}^2$$

We believe these additions give the reader both (a) the concrete physical dimensions of the demonstrated device and (b) a transparent, quantitative view of how the architecture scales to a full matrix.

2. Authors are required to provide uniformity information regarding OTO effect across different matrix unit, such as wavelength tuning linearity for different devices, or power consumption for different devices.

We appreciate the reviewer's request for clarification on the uniformity of the OTO response. To assess this, we fabricated and tested 10 PHIL units with identical parameters. The OTO effect was characterized by quantifying the spectrum shift in response to a calibrated pump laser under identical conditions. The analysis confirms that all devices exhibit highly linear phase tuning with pump power, demonstrating consistent OTO behaviour across the measured devices. The corresponding analysis is now clarified in the main text (line 146) and further detailed in the supplementary section S3.

Main text (line 146)

This responsivity remains highly uniform (5.3% variation) across devices with identical design parameters (Supplementary Fig. S3).

Supplementary Section S2

To evaluate device-to-device uniformity of the OTO effect, ten PHIL units with identical design parameters were fabricated and tested. The OTO response was characterized by quantifying the resonance spectrum shift under excitation from a calibrated pump laser, with all measurements performed under identical conditions (Fig. S3). Across the measured devices, a highly linear dependence of phase shift on pump power was consistently observed, confirming reproducible and uniform OTO behaviour. The extracted phase tuning efficiencies exhibit a mean slope of 0.040

Fig. S3: Uniformity of optical phase tuning across multiple microring devices. (A) Phase tuning response (Δ Phase vs input CS power) for ten PHIL units, each exhibiting a nearly linear thermo-optic response. (B) Statistical summary of the tuning efficiency extracted from (A), showing the mean slope and its standard deviation across all devices ($0.040 \pi/\text{mW} \pm 0.002$, corresponding to 5.3% variation). The small spread confirms highly uniform phase-tuning behaviour within the fabricated array.

π/mW with minimal standard deviation of 0.002π (5.3% variation).

3. What is the overall insertion loss for such design, which can be a crucial parameter for any potential all optical applications?

We thank the reviewer for pointing out this omission. The insertion loss will depend on the wavelength (probe vs control) and the number of nanoheaters employed. For the reported device with 9 nanoheaters on both sides, the insertion loss induced by nanoheaters on the probe signal is ≈ 1.7 dB. A clarification of this point has been included in supplementary section with a detailed analysis of the loss as a function of different device parameters is further.

Supplementary Section S11

The insertion loss (IL) of PHIL devices depends on both wavelength and the number of integrated nanoheaters. To quantify the intrinsic loss of the reported configuration ($R = 30 \mu\text{m}$, 9×2 nanoheaters), devices of identical design were characterized and compared against a nanoheater-free reference fabricated with the same coupler and routing geometry.

Unlike conventional add-drop microrings, where nearly all outcoupled light is collected from a single port, the PHIL design symmetrically couples counter-propagating modes into two ports. In the present setup, only one port is collected, introducing an inherent ~ 3 dB coupling penalty independent of nanoheater absorption.

After accounting for this baseline, the measured IL of the reported PHIL devices is ~ 1.7 dB at the non-absorptive wavelength (λ_{PS}) and ~ 6 dB at the absorptive wavelength (λ_{CS}) (Fig. S15A). The additional attenuation at λ_{CS} arises from localized optical absorption within the nanoheater array, while probe-wavelength losses remain below ~ 2 dB, indicating good transparency and minimal parasitic dissipation.

3D FDTD simulations of a compact PHIL ring ($R = 4 \mu\text{m}$) further confirm the strong dependence of insertion loss on the number of integrated nanoheaters (Fig. S15B). At the absorptive wavelength (λ_{CS}), the IL increases rapidly with heater count, reaching ~ 11.5 dB at $N = 9 \times 2$, reflecting cumulative absorption from the expanding nanoheater array. In contrast, at the non-absorptive wavelength (λ_{PS}), IL remains low (< 2 dB) and scales slowly with N , consistent with minimal modal overlap and negligible absorption. The simulated attenuation contrast therefore increases with heater number, reaching ~ 10 dB at $N = 9 \times 2$, in good agreement with experimental trends. The higher simulated IL compared to measurement (~ 11.5 dB vs. ~ 6 dB) arises mainly from stronger field confinement in the smaller simulated ring and fabrication-related variations that modify the heater-mode overlap.

These results confirm that the primary source of excess loss arises from localized absorption at the selected wavelength. The intrinsic nanoheater-related loss remains low (~ 1.7 dB at the probe wavelength) for the reported 9×2 configuration, demonstrating that the PHIL architecture can achieve strong pump-selective absorption while maintaining low insertion loss, which is an essential feature for scalable, wavelength-multiplexed all-optical modulation and computing.

Fig. S15: Measured and simulated insertion loss of the PHIL architecture. (A) Measured insertion loss of the reported PHIL devices, referenced to identical designs without integrated nanoheaters. The loss at the absorptive wavelength ($\lambda_{\text{absorptive}}$) is significantly higher than that at the non-absorptive wavelength ($\lambda_{\text{non-absorptive}}$), confirming localized optical absorption by the embedded heaters. **(B)** Simulated insertion loss in a compact PHIL configuration as a function of the number of nanoheaters (1×2, 3×2, 5×2, 7×2 and 9×2). Insertion loss increases rapidly with heater count at the absorptive wavelength, reaching ~11.5 dB at N=9×2, due to cumulative optical absorption within the nanoheater array. In contrast, loss at the non-absorptive wavelength remains below 2 dB and scales slowly with heater number, confirming strong wavelength selectivity and minimal parasitic dissipation in the transparent regime.

4. Have authors evaluate the thermal cross-talk issues of nano heaters, especially for large scale integration?

We thank the reviewer for raising the important question regarding thermal crosstalk, particularly in the context of large-scale integration. Our system does not allow for miniaturisation to size and densities which would allow us to measure thermal crosstalk. However, this would become important when integrated in large scale. To understand these dynamics, we have performed thermal simulations which show minimal crosstalk for distances beyond 2 μm . This has been discussed in supplementary section S12. Importantly according to our simulations, the thermal crosstalk is improved compared to standard thermal phase-shifter based designs due to the localisation of heat to the waveguide.

Supplementary section S12:

Finite-difference thermal simulations (*Lumerical HEAT* module) were performed to evaluate the spatial and temporal temperature distribution induced by optical excitation of the PHIL nanoheater array. A 150-ns optical pulse was applied to locally heat the absorptive region. The resulting steady-state temperature profile (Fig. S16) shows a highly localized hotspot centred on the nanoheater array, with a maximum temperature rise of ~ 21 K above ambient. The temperature rapidly decays radially, dropping below 5 K at a distance of ~ 1 μm and below 1 K beyond ~ 2 μm from the heater centre, indicating minimal thermal crosstalk to adjacent photonic components. This localized heating corresponds to an estimated optical phase shift of $\sim 0.025 \pi$ for the reported device configuration, consistent with experimental modulation levels.

Comparison with Conventional Thermo-Optic Phase Shifters:

Unlike standard thermo-optic phase shifters which typically rely on extended metallic heaters^{2,3} (e.g., several tens of microns) and additional driving electronics that introduce parasitic thermal load, our nanoheater design is highly localized (sub-micron scale) and optically actuated, eliminating

Fig. S16: Simulated temperature profile and thermal crosstalk evaluation of PHIL nanoheaters. Transient-state temperature distribution obtained from finite-difference heat simulations (*Lumerical HEAT* module) under a 150-ns optical pulse excitation at the PHIL region. The peak temperature rise is confined within ~ 2 μm around the nanoheater array, indicating minimal thermal crosstalk to adjacent photonic components. Beyond 2 μm separation, the temperature rise falls below 1 K, confirming good thermal isolation suitable for large-scale photonic integration.

the need for electrical contacts. This will significantly reduce the total thermal budget and the spatial extent of heat diffusion. We further note that the standing-wave-enhanced spatial confinement of optical heating in our system intrinsically favours localized thermal interaction with minimal lateral diffusion. Combined with established foundry-compatible strategies for suppressing thermal crosstalk, such as undercutting and thermal isolation trenches, this design supports scalable deployment in densely integrated photonic arrays.

5. Is there feedback control for MRRs? Do authors observe any wavelength drifting over time during CS power sweeping in Fig. 4?

In our implementation, no feedback control was needed. We have further clarified in the manuscript lines 218.

Main text line 218:

To experimentally characterize these effects, we inject a continuous-wave CS at the centre of an absorptive resonance and gradually (\sim Hz) modulate its power over multiple cycles (Supplementary Fig. S12)

The system remains stable with negligible wavelength drift across the entire range of CS power modulation as shown in the Fig.S4 below, which presents the full time-resolved transmission traces, corresponding to the data summarized in the main text (Figure. 4). To further verify the long-term stability, we performed repetitive modulation tests at three probe wavelengths over 1 hour of continuous operation, where the probe transmission follows highly repeatable modulation patterns over extended periods without measurable phase distortion. This was discussed in the supplementary section S8.

Supplementary Section S8:

To characterize the nonlinear response of the PHIL device and assess its resonance stability, we performed repeated modulation tests under open-loop operation, without any active feedback control. The CS power was periodically modulated while monitoring the probe transmission at three resonance wavelengths: 1557.1 nm, 1563.4 nm, and 1569.8 nm.

Fig. S12: Non-linearity measurements over multiple cycles.

Fig. S12 present the full time-resolved transmission traces, corresponding to the data summarized in the main text (Figure. 4). All three channels show highly repeatable thermo-optic modulation with no measurable baseline drift or phase distortion over multiple modulation cycles, confirming the intrinsic thermal stability of the device.

To further assess long-term stability, repetitive modulation was conducted for 1 hour at different wavelengths. The resulting modulation amplitude (ΔT) was analyzed statistically and plotted as mean $\pm \sigma$ in Fig. S13. The small standard deviation ($< \pm 1.7\%$) corresponds to minimal wavelength drift. This long-term consistency further demonstrates the passive thermal robustness of the PHIL structure.

Fig. S13: Long-term (1h) stability of the PHIL device under repeated CS modulation. Statistical analysis of the modulation amplitude (ΔT) as a function of input CS power for the three wavelengths, plotted as mean $\pm \sigma$ over one-hour continuous operation. The minimal deviation ($< \pm 1.7\%$) excellent thermal and spectral stability of the device in open-loop conditions.

6. Would it be possible to implement CS pulse train at higher speed above 100GHz?

This is an important point as the system performance scales with the modulation speed. Although it is theoretically possible to implement a CS pulse train at >100 GHz in our architecture, we do not at present have the experimental capabilities for such an experiment. Modulation frequencies of up to 50 GHz are at the limits of our capabilities. Yet we see no reason why higher frequencies could not be employed.

Reviewer #2:

This paper reports an all-optical computing platform that performs time integration and nonlinear processing using subwavelength thermal nano-heaters embedded in a photonic circuit. The system's ability to apply reconfigurable activation functions is experimentally validated by leveraging thermo-optic effects and standing-wave field patterns. The manuscript can be accepted for publication provided the following questions are adequately addressed:

We thank the reviewer for their insightful feedback and for recommending our manuscript for publication. Below, we provide detailed responses to each point raised, including additional analysis, quantitative benchmarks, and clarifications of our system's design principles. We have revised the manuscript accordingly and believe these additions significantly strengthen the work.

1) Performance of the system should be benchmarked against other all-optical technologies in terms of energy efficiency, speed, and precision.

We appreciate this was not adequately discussed. We have included a detailed benchmarking analysis in the supplementary section S3, accompanied by a comparative table (supplementary Table.S1) summarizing the energy efficiency, speed, and precision of the PHIL device relative to other all-optical technologies.

Supplementary Section S13:

Energy estimation

At 50 GHz repetition rate,

$$T_{\text{pulse}} = \frac{1}{50} \text{GHz} = 20 \text{ps}.$$

Within a 20 ns integration window:

$$N_{\text{pulses}} = 20 \text{ns} / 20 \text{ps} = 1000 \text{pulses}.$$

Assuming half correspond to logical "1" and half to logical "0", approximately 500 active pulses contribute to the energy accumulation.

The nonlinear activation is triggered at an optical power of ≤ 2.6 mW.

Energy per pulse:

$$E_{\text{CS,pulse}} = 52 \text{fJ}.$$

Hence, each "1" pulse costs ~ 52 fJ.

Total control-signal energy per 20 ns window:

$$E_{\text{CS,total}} = 500 \times 52 \text{fJ} = 26000 \text{fJ} = 26 \text{pJ}.$$

Probe-signal contribution (0.1 mW steady across the window):

$$E_{\text{probe}} = 2 \text{pJ}.$$

Therefore, the total energy per integration window is:

$$E_{\text{total}} = E_{\text{CS,total}} + E_{\text{probe}} = 28 \text{pJ}.$$

Normalized Energy per MAC

Each 20 ns integration window accumulates approximately 500 operations (MAC equivalents).

$$E_{\text{per-operation}} = 28\text{pJ}/500 = 0.056\text{pJ} = 56\text{fJ}$$

The PHIL system uniquely performs temporal integration entirely in the optical domain while applying programmable nonlinearity, which is a functionality not achievable by other all-optical technologies. Its opto-thermal design allows summation of thousands of ultrafast optical inputs over a ~ 130 ns thermal window, making it inherently suitable for neuromorphic computing tasks requiring accumulation and thresholding over time.

Compared to phase-change materials (PCM), PHIL is volatile but enables real-time, analogue tunability. PCMs are non-volatile and ideal for static reconfiguration, yet their nanosecond-scale switching hinders high-speed integration. Relative to ultrafast Kerr or 2D material devices, PHIL trades off switching speed for the ability to perform analogue accumulation and nonlinear activation in a single passive platform, leveraging its slow-heat-based integration dynamics.

In summary, PHIL occupies a distinct niche among all-optical integrative computing schemes: it offers repeatable, energy-efficient accumulation with multi-bit analogue precision, balancing performance and tunability where other mechanisms prioritize either speed or non-volatility. While its thermal time constant (~ 130 ns) is slower than sub-picosecond nonlinearities, integration occurs over this window independent of the incoming pulse rate, enabling compatibility with ultrafast (50 GHz) pulse streams and WDM operation, a regime difficult to access using carrier-dynamics-based photonic neuromorphic hardware.

Table S1 | Comparison of All-Optical Light-Controlled Technologies

Technology	Energy per Operation	Speed (Response/BW)	Precision	Optical Temporal Integration + Nonlinear Activation	Ref
Photonic Heater-in-Lightpath (PHIL) Integrator	~56 fJ	~130 ns thermal time constant (≈ 1.2 MHz bandwidth). Can integrate ultrafast (~ 50 GHz) optical pulses over this window	~5-bit analog resolution demonstrated	Yes	This work
Phase-Change Material (PCM)	~tens of pJ	~10–100 ns optical pulse required for phase change	Multi-level analog phase possible (~ 5 -bit) ⁴	No	Li et al. , 2019 ⁴ ; Farmakidis et al. , 2019 ⁵ ; He et al. , 2024 ⁶
Free-Carrier Optical Injection	~hundreds of fJ	~hundreds of ps (carrier lifetime limited)	*	No	Preston et al. , 2008 ⁷ ; Shi et al. , 2022 ⁸
2D Material All-Optical Modulator	~tens of fJ	~hundreds of fs	*	No	Ono et al. , 2020 ⁹
Semiconductor optical amplifier (SOA)	~3 pJ (~100 pJ, if electrical bias included)	~hundreds of ps	Sigmoid activation fitted to logistic function with high agreement (NRMSE < 0.08, dynamic range ~27 dB for cross-connect networks)	Yes-but with electrical bias	Mourgias-Alexandris et al. , 2019 ¹⁰ ; Kravtsov et al. , 2011 ¹¹ ; Shi et al. , 2020 ¹²

* Not reported

2) Thermal crosstalk between neighboring components is not quantified, which could negatively impact the performance. The authors should discuss potential mitigation strategies and scalability constraints.

We thank the reviewer for raising an important point which Reviewer 1 has also raised. This has been further clarified above (Reviewer 1, comment 4). We thank both reviewers for catching an important omission

Thermal crosstalk is a critical consideration for scalable photonic systems. While our current design does not include direct measurements of cross-heater thermal spread, we address this issue from both theoretical and experimental perspectives:

Supplementary section S12:

Finite-difference thermal simulations (*Lumerical HEAT* module) were performed to evaluate the spatial and temporal temperature distribution induced by optical excitation of the PHIL nanoheater array. A 150-ns optical pulse was applied to locally heat the absorptive region. The resulting steady-state temperature profile (Fig. S16) shows a highly localized hotspot centred on the nanoheater array, with a maximum temperature rise of ~ 21 K above ambient. The temperature rapidly decays radially, dropping below 5 K at a distance of ~ 1 μm and below 1 K beyond ~ 2 μm from the heater centre, indicating minimal thermal crosstalk to adjacent photonic components. This localized heating corresponds to an estimated optical phase shift of $\sim 0.025 \pi$ for the reported device configuration, consistent with experimental modulation levels.

Fig. S16: Simulated temperature profile and thermal crosstalk evaluation of PHIL nanoheaters. Transient-state temperature distribution obtained from finite-difference heat simulations (*Lumerical HEAT* module) under a 150-ns optical pulse excitation at the PHIL region. The peak temperature rise is confined within ~ 2 μm around the nanoheater array, indicating minimal thermal crosstalk to adjacent photonic components. Beyond 2 μm separation, the temperature rise falls below 1 K, confirming good thermal isolation suitable for large-scale photonic integration.

Comparison with Conventional Thermo-Optic Phase Shifters:

Unlike standard thermo-optic phase shifters which typically rely on extended metallic heaters^{2,3} (e.g., several tens of microns) and additional driving electronics that introduce parasitic thermal load, our nanoheater design is highly localized (sub-micron scale) and optically actuated, eliminating the need for electrical contacts. This will significantly reduce the total thermal budget and the spatial

extent of heat diffusion. We further note that the standing-wave-enhanced spatial confinement of optical heating in our system intrinsically favours localized thermal interaction with minimal lateral diffusion. Combined with established foundry-compatible strategies for suppressing thermal crosstalk, such as undercutting and thermal isolation trenches, this design supports scalable deployment in densely integrated photonic arrays.

3) The nonlinear activation functions depend on probe wavelength alignment with Lorentzian resonance line shapes. How sensitive is the activation function to fabrication variability or wavelength drift, e.g., due to temperature fluctuations?

We thank the reviewer for this insightful question regarding the sensitivity of the nonlinear activation to resonance alignment and environmental fluctuations. As also discussed in our response to Reviewer 1 (Comment 5) and shown in Supplementary Section S8, the PHIL device demonstrates high spectral and thermal stability during operation even under open-loop conditions with no feedback control.

Experimentally, the device exhibits negligible resonance drift during prolonged operation. As detailed in Supplementary Figures S8, repeated modulation at three probe wavelengths over one-hour results in a standard deviation below $\pm 1.7\%$, corresponding to a wavelength drift of < 0.5 pm, i.e., $< 1\%$ of the resonance linewidth. This minimal fluctuation confirms that the nonlinear activation is insensitive to slow thermal drift under open-air laboratory conditions, demonstrating the intrinsic stability of the PHIL device without environmental isolation or feedback control.

Supplementary Section S8:

To further assess long-term stability, repetitive modulation was conducted for 1 hour at different wavelengths. The resulting modulation amplitude (ΔT) was analyzed statistically and plotted as mean $\pm \sigma$ in Fig. S13. The small standard deviation ($< \pm 1.7\%$) corresponds to minimal wavelength drift. This long-term consistency further demonstrates the passive thermal robustness of the PHIL structure.

Fig. S13: Long-term (1h) stability of the PHIL device under repeated CS modulation. Statistical analysis of the modulation amplitude (ΔT) as a function of input CS power for the three wavelengths, plotted as mean $\pm \sigma$ over one-hour continuous operation. The minimal deviation ($< \pm 1.7\%$) excellent thermal and spectral stability of the device in open-loop conditions.

In practical settings, any larger environmental fluctuation could be compensated with a low-bandwidth feedback loop applied on an extra phase shifter, without interfering with the high-speed optical computation. Established tuning techniques such as integrated thermo-optic phase shifters (TOPS) or local PMMA trimming can be adopted as needed to further enhance alignment precision and device yield.

We would like to thank the reviewer once again for excellent feedback which has considerably strengthened our manuscript.

Reviewer #3:

The manuscript by Zhang et al. presents a novel and compelling approach to all-optical temporal integration and nonlinear activation using a photonic-heater-in-lightpath (PHIL) architecture based on a silicon microring resonator with subwavelength titanium antennas. The work leverages wavelength-selective absorption and thermo-optic effects to achieve simultaneous accumulation and programmable activation, demonstrating potential for scalable neuromorphic photonic computing. The concept is innovative, the device fabrication and experimental validation are thorough, and the results are promising. However, to firmly establish the practical viability and scalability of this approach for real-world AI workloads, the following points require further clarification and quantitative support.

We thank the reviewer for the insightful feedback and for finding our work both innovative and thoroughly demonstrated. Assessing energy efficiency, precision limits, signal integrity, and system-level performance is crucial to validating the scalability of PHIL for neuromorphic computing. Below, we address each point quantitatively and clarify the current experimental scope and future integration potential.

1. The energy consumption per operation is a critical metric for neuromorphic hardware. Please provide the normalized on-chip energy per accumulated MAC operation at a representative vector length. This should include the energy cost of both the control and probe signals, as well as any amplification or conditioning required.

We thank the reviewer for highlighting the importance of benchmarking energy consumption. We have calculated the normalized on-chip energy per accumulated MAC operation. Based on our experimental measurements, where the control signal is modulated at 50 GHz and integrated within the device's thermal window, the effective energy per MAC is **~56 fJ**, which includes both the control and probe signal contributions. The detailed calculation process is specified in the supplementary section S13.

Supplementary Section S13:

Energy estimation

At 50 GHz repetition rate,

$$T_{\text{pulse}} = \frac{1}{50} \text{GHz} = 20 \text{ps}.$$

Within a 20 ns integration window:

$$N_{\text{pulses}} = 20 \text{ns} / 20 \text{ps} = 1000 \text{pulses}.$$

Assuming half correspond to logical "1" and half to logical "0", approximately 500 active pulses contribute to the energy accumulation.

The nonlinear activation is triggered at an optical power of ≤ 2.6 mW.

Energy per pulse:

$$E_{\text{CS,pulse}} = 52 \text{fJ}.$$

Hence, each "1" pulse costs ~52 fJ.

Total control-signal energy per 20 ns window:

$$E_{\text{CS,total}} = 500 \times 52 \text{ fJ} = 26000 \text{ fJ} = 26 \text{ pJ} .$$

Probe-signal contribution (0.1 mW steady across the window):

$$E_{\text{probe}} = 2 \text{ pJ} .$$

Therefore, the total energy per integration window is:

$$E_{\text{total}} = E_{\text{CS,total}} + E_{\text{probe}} = 28 \text{ pJ} .$$

Normalized Energy per MAC

Each 20 ns integration window accumulates approximately 500 operations (MAC equivalents).

$$E_{\text{per-operation}} = 28\text{pJ}/500 = 0.056\text{pJ} = 56 \text{ fJ}$$

For projected large-vector operation (vector length $\sim 250,000$ accumulated within a 130 ns thermal window across 40 WDM channels in one PHIL), the throughput efficiency corresponds to **~ 0.018 TOPS/mW**. Importantly, this estimate is conservative: in practice, under large-scale operation the per-pulse drive can be further optimized or reduced. As such, the effective TOPS/mW performance is expected to improve with system scaling.

2. How does the effective number of bits scale with an increasing number of simultaneous wavelength channels, and what is the primary source of noise or crosstalk that limits the precision in a WDM scenario?

We thank the reviewer for this insightful question regarding precision scaling under wavelength-division multiplexing (WDM) operation. Our initial implementation characterized the effective precision of the PHIL device for single-channel and two-channel WDM operation, as reported in the main text. Following the reviewer's suggestion, we have now extended the analysis to multi-channel WDM, where the effective precision was experimentally evaluated and compared in Supplementary Section S5.

Supplementary section S5:

To quantify the analogue precision of the PHIL device under multi wavelength-division multiplexing (WDM) channels, we evaluate the effective bit resolution (BR) based on the statistical variability of the measured optical response. The dynamic bit resolution is defined as:

$$\text{Bit Resolution (BR)} = \log_2 \left(\frac{1}{\sigma} \right)$$

where σ is the standard deviation of the normalized output error with respect to the ideal additive or integrated response. This metric is equivalent to the effective number of bits (ENOB) in analog systems and serves as a measure of precision for both static and dynamic photonic computation.

1. Precision with a Single WDM Channel

As shown in Figure 3 (main text), we evaluated the bit resolution by quantifying the variability of the integrated output under randomized control-signal (CS) pulse trains. The peak resolution was achieved near the optimal absorption powers ($\sim 1.5\text{--}1.7$ mW), with a mean ENOB of 4.9 bits across the tested power range. This precision reflects the combined influence of photodetector noise, thermal fluctuations, and absorption-induced nonlinearity, indicating stable and reproducible time-domain integration in a single-channel configuration.

2. Precision under Two WDM Channels (Static Modulation)

As shown in Figure 2 (main text), we conducted controlled two-channel WDM summation experiments, where two independent CS wavelengths were simultaneously modulated. The measured output agrees closely with the expected additive response, and the corresponding error histogram yields a standard deviation of $\sigma = 0.0181$ with near-zero mean bias. This corresponds to an effective precision of approximately 5–6 bits, demonstrating the excellent linearity and additive behaviour of the PHIL device under static multi-wavelength excitation.

3. Precision under Three WDM Channels (Static Modulation)

To examine the scalability of precision with additional multiplexed inputs, the WDM summation experiment was extended to a three-wavelength configuration (Supplementary Fig. S7). The resulting error histograms (Fig. S7C) show a standard deviation of $\sigma \approx 0.04$, corresponding to an effective resolution of ~ 4.5 bits, which is slightly lower than the 5–6 bits observed in the two-channel case. The modest reduction arises from shared thermal loading and additional summation noise introduced by the extra wavelength path. Despite this, the retained >4 -bit precision confirms accurate multi-channel optical accumulation under concurrent WDM excitation.

Importantly, the device response remains highly linear across all tested configurations, and the bit precision lies well within the usable analogue range for neuromorphic photonic computation. We anticipate that improved thermal isolation, optimized nanoheater geometries, and balanced input-power normalization will further mitigate inter-channel coupling in future device generations, enabling higher-bit operation under large-scale WDM operation.

Fig. S7: All-optical summation across 3 wavelengths. (A). Measured nonlinear summation surface of the probe transmission (ΔT) as a function of three independent control-signal (CS) powers (a, b, c) at distinct. The probe output increases monotonically with total optical power, confirming additive thermal accumulation across multiple wavelengths. (B). Comparison between the measured output and the ideal additive response, showing excellent linear correlation. (C). Histogram of the deviation between measured and expected responses, yielding a standard deviation of $\sigma = 0.0437$ (mean = 0.0162), corresponding to an effective precision of approximately **4.5 bits**. These results demonstrate that the PHIL architecture maintains high linearity and low crosstalk even under simultaneous multi-wavelength excitation.

3. Can a multi-stage cascade be demonstrated using only optical probe signals for interconnects, and what is the required per-stage link budget and optical signal-to-noise ratio to ensure the nonlinear activation function remains stable under the cumulative effect of thermal loading and resonance shifts?

An excellent point raised by the reviewer. In the present work, we have experimentally demonstrated single-stage accumulation and activation, where the probe signal is modulated purely by optical control (CS) and exhibits stable, monotonic responses under various power patterns. While a full end-to-end multi-stage cascade is beyond the current experimental scope, a cascade using only optical probe signals as interconnects is fully feasible in the PHIL framework.

In this scheme, the probe output of stage i directly drives the probe input of stage $i + 1$ at the same wavelength, while each stage employs its own local CS wavelengths for nonlinear activation.

1. Per-stage link budget (fixed at 2 dB IL).

Measured probe insertion losses and routing overheads justify a conservative 2 dB per stage. The probe power after M stages is

$$P_M = P_0 10^{-0.2M}$$

Given a minimum probe level P_{min} , the gain-less cascade depth is

$$M_{max} = \left\lceil \frac{10}{2} \log_{10} \left(\frac{P_0}{P_{min}} \right) \right\rceil = \left\lceil 5 \log_{10} \left(\frac{P_0}{P_{min}} \right) \right\rceil$$

With $P_0 = 1.0 \text{ mW}$, and $P_{min} = 0.1 \text{ mW}$, we obtain $M_{max} \approx 5$. Additional stages can be supported by inserting low-noise on-chip gain (or modestly increasing P_0) between blocks.

2. OSNR target for stable nonlinearity across M stages.

To preserve a ≥ 5 -bit effective resolution end-to-end (≈ 32 dB electrical SNR) across M stages, the optical signal-to-noise ratio must satisfy^{13,14}

$$\text{OSNR} \gtrsim 32 + 10 \log_{10} (B_{det}/0.1 \text{ nm}) \text{ dB}$$

where B_{det} is the detection bandwidth.

For a typical thermal bandwidth of 1–2 MHz ($B_{det} \approx 10^{-3} \text{ nm}$), this corresponds to an OSNR $\gtrsim 25$ –30 dB per 0.1 nm, readily met by commercial narrow-linewidth lasers¹⁵. This ensures that cumulative amplitude noise and minor resonance drifts remain below the 1 % fluctuation limit associated with the measured 5-bit precision ($\sigma \approx 0.02$).

3. Thermal loading and resonance-shift control.

Cascaded stability is maintained by controlling each stage within the demonstrated $\leq \sim 2 \text{ mW}$ linear CS window, keeping the net detuning per stage a small fraction of the linewidth ($\approx 10\%$ FWHM guard band), and reducing thermo-optic crosstalk via heater layout optimization, thermal isolation trenches¹⁶.

Overall, a probe-only cascade is directly compatible with PHIL. With 2 dB/stage insertion loss, 3–5 passive stages are achievable without gain at typical probe levels, while deeper cascades are enabled by sparse, low-noise amplification.

4. How do the fundamental constraints imposed by the Kramers-Kronig relations between absorption and phase shift limit the maximum achievable differential responsivity without introducing unacceptable levels of optical loss or distortion for the probe wavelength?

We thank the reviewer for this fundamental question regarding the Kramers–Kronig (K–K) constraints linking absorption and refractive-index modulation, and their implications for the maximum achievable differential responsivity in our device.

In the PHIL architecture, we do not rely on electronic dispersion near the probe frequency to induce refractive-index change. Instead, we absorb control light (CS) spectrally detuned from the probe in subwavelength metallic nanoheaters, where the absorbed optical energy is converted to heat and subsequently drives the thermo-optic phase shift of the probe. This approach replaces direct absorptive dispersion with a thermally mediated refractive index change. The strength of the TO effect is therefore bound by the material.

Moreover, standing-wave engineering positions the probe near an electric-field minimum at the heater locations, ensuring negligible optical field overlap with the absorptive region. Consequently, the K–K coupling that would otherwise impose additional probe loss is largely decoupled from the operating band. In this regime, the limiting mechanisms for responsivity are thermal efficiency and thermo-optic crosstalk, rather than probe-band absorption or dispersive distortion.

To maintain high differential responsivity without introducing measurable probe loss or line-shape distortion, three design constraints are enforced:

1. Spectral guard band: the control-signal (CS) absorption band is detuned from the probe wavelength.
2. Spatial decoupling: the heater-probe overlap is minimized via standing-wave placement, preserving the effective $dT/d\phi$ while minimizing parasitic attenuation.
3. Operational guard band: the total resonance shift per operation is limited to $\leq 10\%$ of the resonance FWHM, preventing distortion of the Lorentzian response.

Under these combined constraints, the PHIL device achieves large thermo-optic responsivity without incurring unacceptable probe loss or nonlinear distortion. This is experimentally verified through the stable, repeatable transfer characteristics and ≈ 5 -bit effective precision observed across both two-channel WDM and time-domain integration measurements.

5. Can the authors demonstrate the device performing an end-to-end neuromorphic inference task such as MNIST handwritten digit recognition, and report how the achieved accuracy compares against a state-of-the-art electronic baseline?

We thank the reviewer for this important question concerning the demonstration of end-to-end neuromorphic inference using the PHIL architecture. The current work focuses on establishing the physical basis of the photonic-heater-in-lightpath (PHIL) concept, specifically, demonstrating analogue temporal integration, nonlinear activation, and multi-wavelength summation within a single integrated photonic element. These operations constitute the core primitives required for scalable optical neuromorphic computation.

While a full on-chip inference pipeline is beyond the scope of the work and our experimental capabilities, we have performed numerical network-level simulations using the experimentally measured PHIL transfer characteristics. The measured nonlinear activation function was directly incorporated into a PyTorch-trained feed-forward neural network, replacing the standard electronic ReLU function. When tested on the MNIST handwritten digit recognition task, the PHIL-based model achieved a classification accuracy of 97.2 %, compared with 97.8 % for the electronic ReLU baseline.

This simulation demonstrates that the experimentally validated PHIL activation function can be seamlessly integrated into standard training workflows and deliver inference accuracy comparable to conventional digital networks. A fully photonic inference demonstration, leveraging on-chip cascaded PHIL neurons with optical fan-out and reconfigurable weighting, is currently under development and will be reported in future work.

Supplementary Section S7:

The experimentally measured PHIL transfer curve was normalized to the range $[-0.2, 1.0]$ and implemented as a piecewise-linear differentiable activation in PyTorch (Fig. S11a). The activation was integrated into a fully connected network (784–256–128–10) for the MNIST handwritten-digit classification task, replacing the standard ReLU function (Fig. S11b). Training used the Adam optimizer ($\text{lr} = 3 \times 10^{-4}$, 20 epochs, batch = 64) with adaptive rescaling to match the PHIL input range. The dataset was pre-normalized to $[-0.2, 1.0]$.

The PHIL-activated network achieved a test accuracy of 97.2 %, compared with 97.8 % for ReLU (Figs. S11c–e), showing nearly identical convergence and uniform class accuracy. These results confirm that the experimentally realized PHIL nonlinearity provides sufficient precision and monotonicity for neuromorphic inference, achieving electronic-level performance while being derived entirely from an optical thermo-optic mechanism.

Fig. S11: Simulation of PHIL activation function in a neural network for handwritten digit recognition. (A). Experimentally measured PHIL transfer curve (red circles) and interpolated nonlinear activation function (blue line) used for the network simulation. (B). Schematic of the fully connected neural network architecture (784–256–128–10) used for MNIST handwritten digit classification, where the experimental PHIL activation replaces the standard electronic ReLU function. (C). Confusion matrix of the PHIL-activated network showing accurate digit recognition across all classes with classification accuracy of **97.2 %**, compared to **97.8 %** for the ReLU baseline. (D). Training loss and validation accuracy versus epoch for PHIL and ReLU activations, demonstrating nearly identical convergence behaviour.

Reference:

1. Giamougiannis, G. *et al.* Neuromorphic silicon photonics with 50 GHz tiled matrix multiplication for deep-learning applications. *AP* **5**, 016004 (2023).
2. Harris, N. C. *et al.* Efficient, compact and low loss thermo-optic phase shifter in silicon. *Opt. Express, OE* **22**, 10487–10493 (2014).
3. Jacques, M. *et al.* Optimization of thermo-optic phase-shifter design and mitigation of thermal crosstalk on the SOI platform. *Opt. Express, OE* **27**, 10456–10471 (2019).
4. Li, X. *et al.* Fast and reliable storage using a 5 bit, nonvolatile photonic memory cell. *Optica, OPTICA* **6**, 1–6 (2019).
5. Farmakidis, N. *et al.* Plasmonic nanogap enhanced phase-change devices with dual electrical-optical functionality. *Science Advances* **5**, eaaw2687 (2019).
6. He, Y. *et al.* Energy-Efficient Integrated Electro-Optic Memristors. *Nano Lett.* **24**, 16325–16332 (2024).
7. Preston, K., Dong, P., Schmidt, B. & Lipson, M. High-speed all-optical modulation using polycrystalline silicon microring resonators. *Applied Physics Letters* **92**, 151104 (2008).
8. Shi, Y. *et al.* Nonlinear germanium-silicon photodiode for activation and monitoring in photonic neuromorphic networks. *Nat Commun* **13**, 6048 (2022).
9. Ono, M. *et al.* Ultrafast and energy-efficient all-optical switching with graphene-loaded deep-subwavelength plasmonic waveguides. *Nat. Photonics* **14**, 37–43 (2020).
10. Mourgias-Alexandris, G. *et al.* An all-optical neuron with sigmoid activation function. *Opt. Express, OE* **27**, 9620–9630 (2019).
11. Kravtsov, K., Fok, M. P., Prucnal, P. & Rosenbluth, D. Ultrafast All-Optical Implementation of a Leaky Integrate-and-Fire Neuron. *Optics express* **19**, 2133–47 (2011).
12. Shi, B., Calabretta, N. & Stabile, R. Deep Neural Network Through an InP SOA-Based Photonic Integrated Cross-Connect. *IEEE Journal of Selected Topics in Quantum Electronics* **26**, 1–11 (2020).

13. Goodman, J. W. *Introduction to Fourier Optics*. (Roberts and Company Publishers, 2005).
14. IEEE Standard for Terminology and Test Methods for Analog-to-Digital Converters. *IEEE Std 1241-2010 (Revision of IEEE Std 1241-2000)* 1–139 (2011) doi:10.1109/IEEESTD.2011.5692956.
15. Dong, P. *et al.* Thermally tunable silicon racetrack resonators with ultralow tuning power. *Opt. Express, OE* **18**, 20298–20304 (2010).
16. Ceccarelli, F. *et al.* Low Power Reconfigurability and Reduced Crosstalk in Integrated Photonic Circuits Fabricated by Femtosecond Laser Micromachining. *Laser & Photonics Reviews* **14**, 2000024 (2020).